# Histone H2B ubiquitylation represses gametogenesis by opposing RSC-dependent chromatin remodeling at the *ste11* master regulator locus

Philippe Materne[1], Enrique Vázquez[2], Mar Sánchez[2], Carlo Yague-Sanz[1], Jayamani Anandhakumar[1†], Valerie Migeot[1], Francisco Antequera[2], Damien Hermand[1*]

[1]URPHYM-GEMO, Namur Research College, University of Namur, Namur, Belgium; [2]Instituto de Biología Funcional y Genómica, Salamanca, Spain

**Abstract** In fission yeast, the *ste11* gene encodes the master regulator initiating the switch from vegetative growth to gametogenesis. In a previous paper, we showed that the methylation of H3K4 and consequent promoter nucleosome deacetylation repress *ste11* induction and cell differentiation (*Materne et al., 2015*) but the regulatory steps remain poorly understood. Here we report a genetic screen that highlighted H2B deubiquitylation and the RSC remodeling complex as activators of *ste11* expression. Mechanistic analyses revealed more complex, opposite roles of H2Bubi at the promoter where it represses expression, and over the transcribed region where it sustains it. By promoting H3K4 methylation at the promoter, H2Bubi initiates the deacetylation process, which decreases chromatin remodeling by RSC. Upon induction, this process is reversed and efficient NDR (nucleosome depleted region) formation leads to high expression. Therefore, H2Bubi represses gametogenesis by opposing the recruitment of RSC at the promoter of the master regulator *ste11* gene.

*For correspondence: Damien. Hermand@unamur.be

**Present address:** †Department of Biochemistry & Molecular Biology, LSU Health Sciences Center, Shreveport, United States

**Competing interests:** The authors declare that no competing interests exist.

## Introduction

The RNA polymerase II (PolII) subunit Rpb1 C-terminal Domain shows a stereotypical pattern of CTD phosphorylation with phospho-S5 (S5P) peaking near the transcription start site (TSS) and phospho-S2 (S2P) accumulating towards the 3'-end of the transcribed region (*Buratowski, 2009*; *Cassart et al., 2012*; *Drogat and Hermand, 2012*). The distribution of histone H3 lysine 4 methylation (H3K4me) mirrors S5P due to the direct recruitment of the H3 methyltransferase Set1-COMPASS by the S5P of PolII (*Keogh et al., 2005*; *Ng et al., 2003*). In budding yeast, Set1 is the only H3K4 methyltransferase and plays a repressive role on *PHO5*, *PHO84* and *GAL1* expression, suggesting that H3K4me creates a repressive chromatin configuration (*Carvin and Kladde, 2004*; *Wang et al., 2011*). Most genes are also upregulated in fission yeast when Set1 is absent (*Lorenz et al., 2014*). The PHD finger protein Set3, which is part of the SET3C HDAC complex, binds H3K4me2 to mediate deacetylation of histones in the 5' regions (*Kim and Buratowski, 2009*; *Kim et al., 2012*). Similarly, the PHD domain of the HDAC-associated ING2 protein mediates its binding to the di- and trimethylated H3K4 at the promoters of proliferation genes (*Pena et al., 2006*; *Shi et al., 2006*).

Histone H2B is monoubiquitylated (H2Bubi) on the conserved lysine 119 in fission yeast. In yeast, Rad6 and Bre1 function as conjugating enzyme (E2) or ligase (E3) (*Hwang et al., 2003*; *Robzyk et al., 2000*; *Wood et al., 2003*). The pathway is conserved in fission yeast with the Rhp6

E2 and two Bre1 homologues, Brl1 and Brl2, a situation closer to higher eukaryotes (*Tanny et al., 2007*; *Zofall and Grewal, 2007*). The presence of the Ubp8 deubiquitylase, which is part of the SAGA co-activator complex, underlies the dynamic nature of H2Bubi (*Daniel et al., 2004*; *Henry et al., 2003*). A second deubiquitylase, Ubp10, modulates the pool of H2Bubi (*Emre et al., 2005*; *Gardner et al., 2005*). H2Bubi functions in a trans-tail regulation of H3K4 and H3K79 methylation (*Briggs et al., 2002*; *Dover et al., 2002*; *Ng et al., 2002*; *Sun and Allis, 2002*). The Set1-COMPASS subunit Swd2 is required for the crosstalk by mechanisms implying its direct ubiquitylation by Rad6-Bre1 (*Vitaliano-Prunier et al., 2008*) or its recruitment by H2B-ub1 (*Lee et al., 2007*). H2Bubi is spread uniformly across transcribed units and at promoters (*Batta et al., 2011*; *Schulze et al., 2011*; *2009*). Ubp8 acts at the 5' region where H3K4me3 is also high while Ubp10 targets the H3K79me3 decorated nucleosomes more typical of 3' regions, which suggests different, maybe opposite, roles of H2Bubi over the length of genes and spatial regulation (*Schulze et al., 2011*). Nucleosome occupancy (*Batta et al., 2011*) in a strain lacking H2Bubi revealed a role in promoting nucleosome assembly leading to repression at promoters and a positive role during elongation by facilitating the eviction of the H2A-H2B dimer and nucleosome reassembly following the passage of the polymerase. However, how H2Bubi represses transcription at the promoter is not clear.

From the SWI/SNF-class remodeling complexes, RSC is ten-fold more abundant and is essential for viability, in contrast the SWI/SNF complex (*Cairns et al., 1996*). RSC is required for promoter nucleosome location (*Hartley and Madhani, 2009*), consistent with its ability to slide nucleosomes in vitro (*Lorch et al., 2011*) and its global requirement for RNA polymerase II transcription (*Parnell et al., 2008*). RSC recognizes acetylated nucleosomes through tandem bromodomains (*Carey et al., 2006*; *Kasten et al., 2004*), which links its recruitment to acetylation of histone H3 lysine 14.

Here we show that promoter H2B ubiquitylation represses the expression of the master regulator of gametogenesis *Ste11* in fission yeast (*Anandhakumar et al., 2013*) by promoting Set1/H3K4me dependent deacetylation, which impedes the recruitment of the RSC complex. A H2B K119R mutant results in decreased nucleosome occupancy at the *ste11* promoter and derepression of the gene, while the absence of RSC has the opposite effect. Therefore, the opposing role of promoter histone H2B ubiquitylation and RSC-dependent chromatin remodeling controls gametogenesis in fission yeast.

## Results

### The abolition of histone H2B ubiquitylation suppresses the *lsk1* deletion mutant

In fission yeast, the CTD S2 kinase Lsk1 (l̲atrunculin s̲ensitive k̲inase - Cdk12 in higher eukaryotes) is required for the completion of cytokinesis in response to perturbation of the actomyosin ring by latrunculin A (LatA) (*Karagiannis et al., 2005*), and the deletion of *lsk1* results in sensitivity to LatA. It is therefore likely that the efficient transcription of one or several genes controlling cytokinesis requires S2P. Testing the LatA sensitivity of the genes we have previously identified (*Coudreuse et al., 2010*) as downregulated in a S2A mutant identified the essential SIN (Septation Initiation Network) component encoding gene *cdc14* (*Fankhauser and Simanis, 1993*) as a potential transcriptional target of Lsk1. The overexpression of *cdc14* significantly suppresses the LatA sensitivity of a strain deleted for *lsk1* (*Figure 1—figure supplement 1A,B*), supporting that the LatA sensitivity of the *lsk1* mutant relates to the downregulation of *cdc14*.

We reasoned that the easily tested LatA sensitivity phenotype should allow us to identify additional gene products acting together with S2P to control gene expression. We therefore screened the entire *S. pombe* deletion library for sensitivity to the presence of 0.5 µM LatA. The screen is not expected to be very specific as many regulators of the actomyosin ring and cytokinesis were already shown to be sensitive to LatA but transcriptional regulators should easily be extracted from that group based on annotation. The sensitivity of the S2A mutation within the CTD is specific as an S7A mutant behaves as wild type (*Figure 1—figure supplement 1C*). The screening appeared to be consistent as the three kinases (Pmk1, Pek1 and Mkh1) forming the MAPK cell integrity transduction pathway were isolated. In addition, we identified the SAGA-associated Ubp8 H2B ubiquitin protease

and several components of the RSC and SWI/SNF chromatin remodeling complexes. We next concentrated on the possible interplay of these transcriptional regulators with S2P.

In order to test a genetic link between S2P and the H2Bubi pathway, we generated double *lsk1 ubp8* and *lsk1 htb1 K119R (htb1* is the single fission yeast gene encoding histone H2B and K119 the target site for ubiquitylation) mutant strains. Surprisingly, the *htb1 K119R* mutant completely suppressed the *lsk1* sensitivity to the drug (*Figure 1A*), contrary to the deletion of *ubp8*.

We next tested if this strong genetic link identified between S2P and H2Bubi was effective at the *cdc14* locus. The decrease in the level of the *cdc14* mRNA observed in the *lsk1* mutant was indeed suppressed when H2Bubi was abolished in the *htb1 K119R* mutant while the *ubp8* mutant had no effect (*Figure 1B*).

The *ste11* gene is another key target of S2P (*Coudreuse et al., 2010*; *Materne et al., 2015*; *Schwer et al., 2012*; *Sukegawa et al., 2011*), which is unrelated to LatA sensitivity. Lsk1 and S2P are critical regulators of the induction of gametogenesis by *Ste11* by releasing Set1/H3K4me repression at the *ste11* locus (*Materne et al., 2015*). Similarly to *cdc14*, we observed that the abolition of H2Bubi led to an increase of *ste11*, including at the basal, non-induced state, supporting that ubiquitylation of H2B represses *ste11* expression (*Figure 1C*). This was clearly an effect of the absence of ubiquitin on H2B as an *rhp6* deletion mutant (lacking the E2 ubiquitin conjugating enzyme) behaved similarly (*Figure 1—figure supplement 1D*). Paradoxically, the *ubp8* deletion behaved similarly to the *htb1 K119R* mutant, also resulting in an increased level of *ste11* (*Figure 1C*). Therefore, either the absence, or the constitutive presence of ubiquitin on H2B results in the derepression of *ste11*. However, only the *htb1 K119R* mutant suppressed the defect of expression manifested in the *lsk1* deletion strain, in a way reminiscent of the *cdc14* case (*Figure 1C*), which was confirmed by a phenotype assay (exposure to iodine that highlights gametogenesis by staining the gametes dark) (*Figure 1D*).

These genetic data suggest that the essential function of Lsk1 (namely S2P) for *cdc14* or *ste11* expression is not required when H2B cannot be ubiquitylated. Contrary, Lsk1 function is required for *ubp8* to upregulate *ste11* expression.

## Histone H2B ubiquitylation is spatially modulated at the *ste11* locus

H2B ubiquitylation was previously analysed in fission yeast using a flag-tagged version (*Sanso et al., 2012*; *Zofall and Grewal, 2007*). We tested the specificity of a new monoclonal antibody in fission yeast to avoid the caveats of using a tagged version. Despite two amino acid changes in the predicted epitope (*Figure 1—figure supplement 2A*), the antibody specifically recognized the ubiquitylated version of H2B as no signal was observed in a K119R mutant (*Figure 1—figure supplement 2B*). Humanizing the fission yeast gene (*Figure 1—figure supplement 2C*) in order to generate the exact epitope did not improve recognition, indicating that the antibody could be used on a wild type strain. The H2Bubi signal appeared increased in the *ubp8* deletion strain but not in the absence of *ubp16* that encodes the closest fission yeast orthologue to *ScUBP10* (*Kouranti et al., 2010*; *Sadeghi et al., 2014*). The double *ubp8 ubp16* mutant had similar level of H2Bubi to the single *ubp8* mutant, suggesting that Ubp8 is the genuine H2B deubiquitylase in fission yeast as previously proposed (*Sadeghi et al., 2014*) (*Figure 1—figure supplement 2B*).

In order to decipher the complex genetic interactions between S2P and H2Bubi, we followed the spatial distribution of H2B ubiquitylation over the *ste11* locus during induction. A reverse pattern was observed at the promoter and 3'-distal region with H2Bubi occupancy decreasing at the promoter and increasing along the transcribed unit during gene induction (*Figure 1E*, left panel). Considering the dependency of H3K4me to H2Bubi, we also followed the trimethyl mark (H3K4me3) that behaved similarly to H2Bubi at the promoter but was very low in the 3' region of *ste11* (*Figure 1E*, right panel). Therefore, the use of the *htb1 K119R* mutant confirmed that the deposition of H3K4me3 requires previous H2B ubiquitylation (*Figure 1E*, right panel). We reasoned that the opposite dynamic of H2Bubi at the promoter and 3' region could reflect that the ubiquitylation represses transcription at the promoter while it promotes it during elongation, which could explain why both the *htb1 K119R* and the *ubp8* deletion mutant lead to increased expression of *ste11*. In addition, our recent work indicated that H3K4me3 represses *ste11* expression by recruiting the SET3C HDAC, which is counteracted by Lsk1-dependent S2P (*Materne et al., 2015*). Therefore, in the *htb1 K119R* strain where H3K4me3 is downregulated (*Figure 1E*, right panel), Lsk1 and S2P would not be required, which is what we observe both genetically (*Figure 1D*) and by RT-Q-PCR

Materne et al. eLife 2016;5:e13500. DOI: 10.7554/eLife.13500

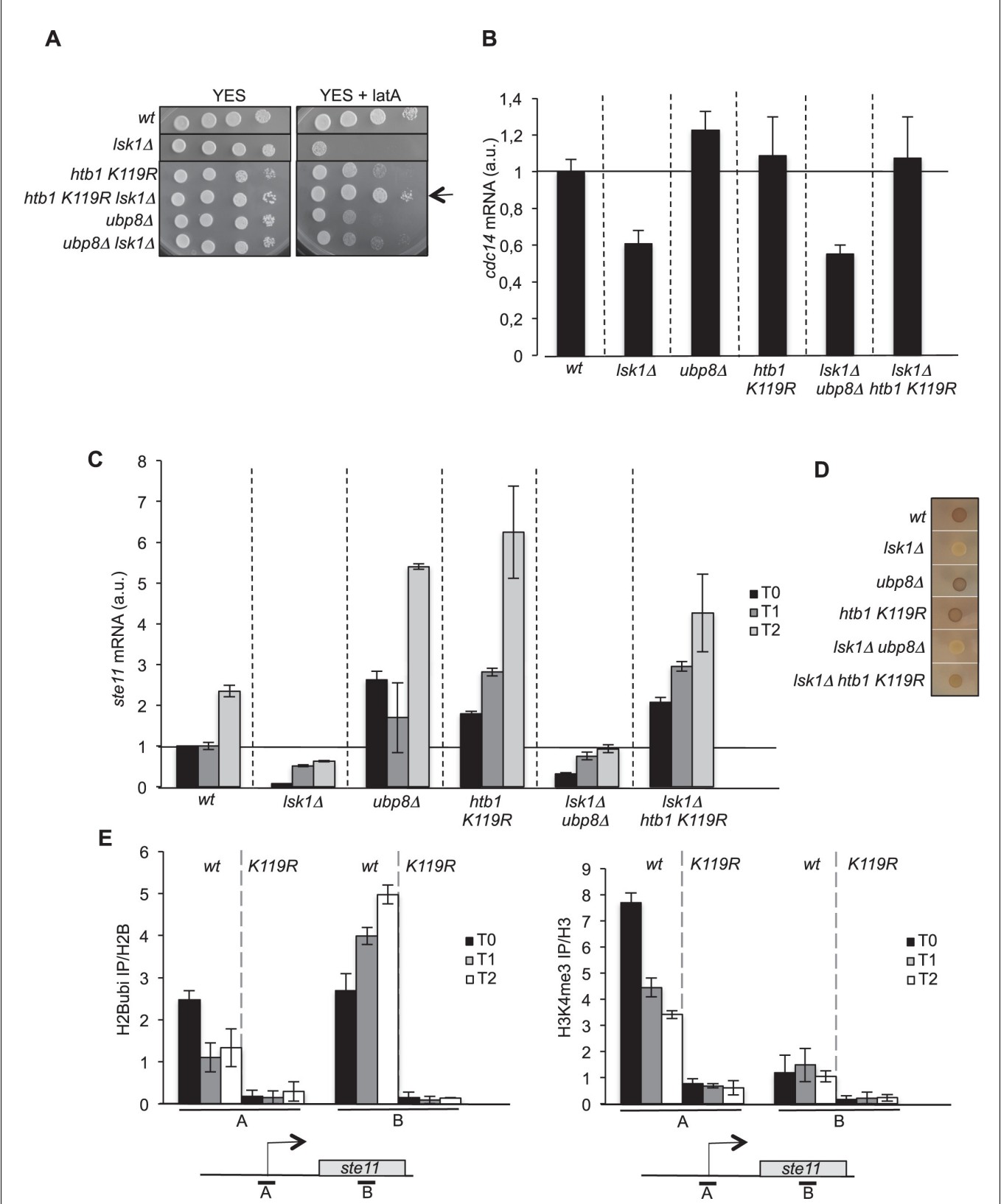

**Figure 1.** The abolition of histone H2B ubiquitylation suppresses the requirement of *lsk1*. (**A**) Spot dilution assay of indicated strains grown 2 days at 32°C on rich medium in the presence or absence of LatA (0.5 µM). The arrow indicates the suppression of *lsk1* growth defect by the *htb1 KR* mutant.
*Figure 1 continued on next page*

*Figure 1 continued*

(B) Relative quantification of the *cdc14* mRNA determined by quantitative RT-Q-PCR in the indicated strains. a.u.: arbitrary units. Each column represents the averaged value ± SEM (n = 3). (C) Relative quantification of the *ste11* mRNA determined by quantitative RT-Q-PCR in the indicated strains during vegetative growth (T0) and nitrogen starvation at the indicated time points (hours). a.u.: arbitrary units. Each column represents the averaged value ± SEM (n = 3). (D) The indicated strains were plated for 48 hr on mating medium (malt extract) before iodine staining to reveal sterility. (E) Left panel. The *wt* and *htb1 K119R* strains were starved for nitrogen at the indicated time points (hours). The occupancy of ubiquitylated H2B at the indicated locations (A, B) was determined by ChIP using the anti-H2Bubi normalized against unmodified H2B. Right panel. The *wt* and *htb1 K119R* strains were starved for nitrogen at the indicated time points (hours). The occupancy of ubiquitylated H2B at the indicated locations (A, B) was determined by ChIP using the anti-H3K4me3 normalized against unmodified H3. The same chromatin sample was used for the left and right panel. Each column represents the averaged value ± SEM (n = 3)

The following figure supplements are available for figure 1:

**Figure supplement 1.** The overexpression of *cdc14* rescues the LatA sensitivity of a *lsk1* deleted strain.

**Figure supplement 2.** Characterization of the histone H2B mutants generated.

(*Figure 1C*). We further tested this model by measuring the level of H3 at the promoter of various strains. Confirming our previous data, the deletion of *lsk1* or the mutation of H3 lysine 14 to arginine (Htt2 K14R) led to an elevated level of H3 onto the promoter region while the *htb1 K119R* mutant, or a strain lacking *hos2* (encoding the catalytic subunit of the SET3C HDAC) had a reduction of histone occupancy at the *ste11* promoter (*Figure 2A*). Importantly, the level of histone observed in the absence of *ubp8* was very similar to the wild type, indicating that the upregulation of *ste11* expression proceeds through a different mechanism, as already suggested above.

We conclude that H2B ubiquitylation at the promoter of *ste11* correlates with repressed transcription and high level of H3K4me3, which in turn may favour histone deacetylation by the SET3C complex. However, it is unclear how this pathway and the increased occupancy of deacetylated nucleosomes at the *ste11* promoter are counteracted upon gene induction. We next tested the involvement of components of the RSC and SWI/SNF remodeling complexes identified in the LatA sensitivity screen (*Figure 1—figure supplement 1C*) in the expression of the *ste11* and *cdc14* genes.

## The RSC remodeling complex is required to establish a large NDR at the *ste11* promoter

The RSC and the SWI-SNF complexes share 6 subunits (*Figure 2—figure supplement 1A*). We first analyzed the effect of the deletion of *rsc1* (RSC-specific subunit), *snf22* (SWI/SNF-specific catalytic subunit, the orthologue of budding yeast Snf2) and *arp9* (shared subunit) on the induction of *ste11*. The absence of either *rsc1* or *arp9* clearly affected the expression of *ste11* while *snf22* had no effect (*Figure 2B*), pointing to a more prominent role of RSC for *ste11* expression, which was confirmed by a gametogenesis assay (*Figure 2—figure supplement 1B*). In the case of *cdc14*, gene expression was affected by both the RSC and the SWI/SNF complexes (*Figure 2—figure supplement 1C*) as already anticipated from the identification of both *rsc1* and *snf22* in the LatA screen (*Figure 1—figure supplement 1C*). The gene specificity in the requirement of either chromatin remodelers was supported by a ChIP experiment indicating that the SWI-SNF specific subunit Snf22 is present at the *cdc14* promoter but not at the *ste11* promoter (*Figure 2—figure supplement 1D*). In order to analyze the effect of the essential catalytic subunit of RSC, Snf21, on *ste11*, we generated a switch off system based on the rTetR-TetO system (*Zilio et al., 2012*) where the addition of anhydrotetracycline (ahTet) leads to the transcriptional repression of *snf21* (*Figure 2—figure supplement 1E*). *Figure 2C–D* shows that the level of the Snf21 protein is strongly downregulated and cell viability drops in the presence of ahTet. Within 3 hr of growth in the presence of the drug, the level of both the *snf21* and *ste11* mRNAs was decreased by 60% (*Figure 2E–F*). Moreover, within the time scale of the depletion of *snf21*, the level of histone H3 increased at the *ste11* promoter (*Figure 2G*).

These data indicate that the catalytic subunit of RSC, is required for *ste11* expression, most likely through the displacement of nucleosomes. In order to confirm this possibility, we next analyzed the level of H3 at the promoter of *ste11* in a strain lacking *arp9* (*Figure 3A*), which revealed a marked

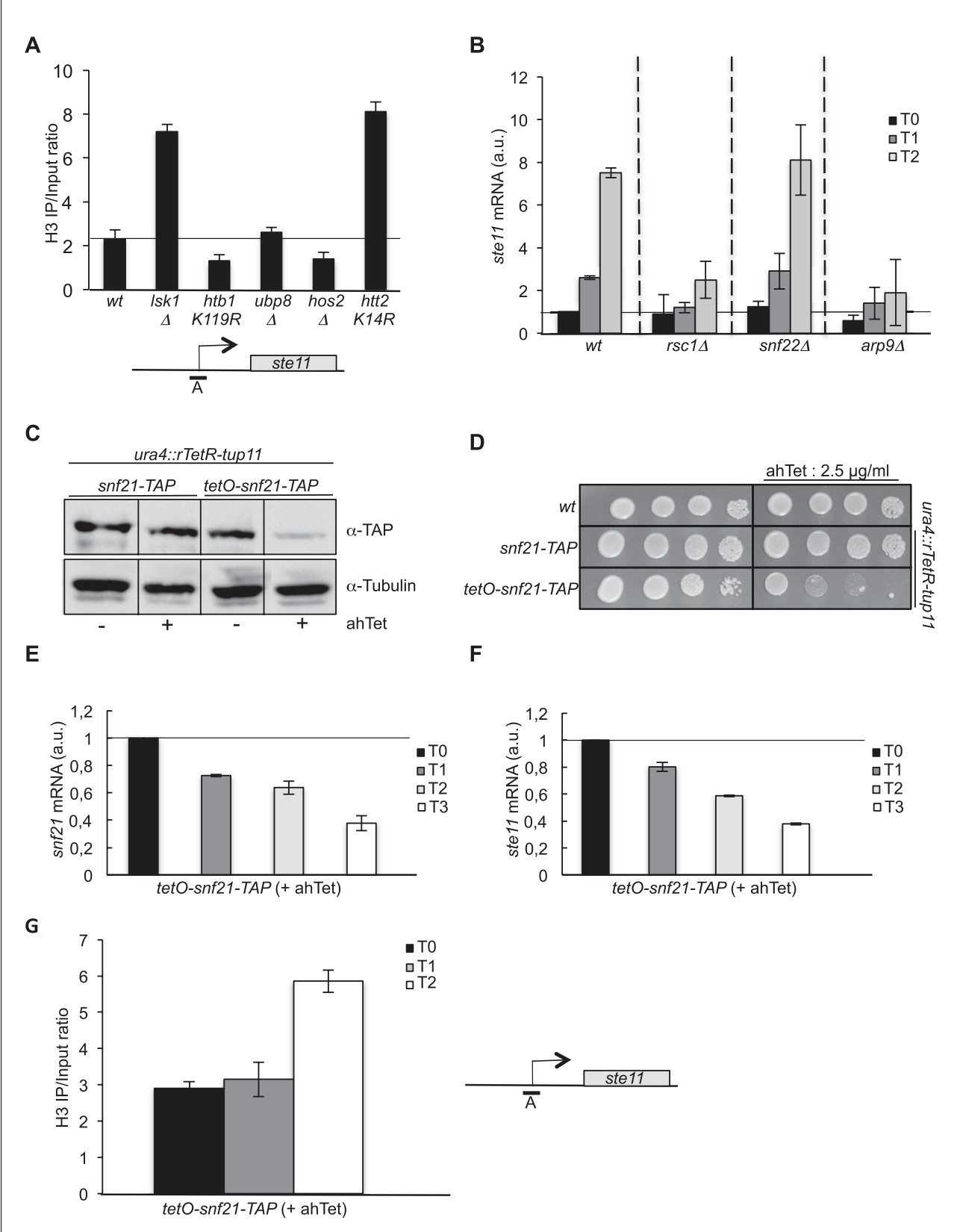

**Figure 2.** The RSC complex is required for the induction of *ste11*. (**A**) The occupancy of histone H3 at the *ste11* promoter was measured by ChIP using the indicated amplicon in the indicated strains. Each column represents the averaged value ± SEM (n = 3). (**B**) Relative quantification of the *ste11* mRNA

*Figure 2 continued on next page*

*Figure 2 continued*
determined by quantitative RT-Q-PCR in the indicated strains during vegetative growth (T0) and nitrogen starvation at the indicated time points (hours). a.u.: arbitrary units. Each column represents the averaged value ± SEM (n = 3). (C) Western blot analysis (anti-TAP and anti-tubulin) of total protein extracts from the indicated strains grown 2 hr in the presence or absence of anhydrotetracycline (ahTet 2.5 µg/ml). (D) Spot dilution assay of indicated strains grown 2 days at 32°C on rich medium in the presence or absence of ahTet (2.5 µg/ml). (E) Relative quantification of the *snf21* mRNA determined by quantitative RT-Q-PCR in the indicated strain grown in the presence of ahTet for three hours. Samples were taken at the indicated time (hours). a.u.: arbitrary units. Each column represents the averaged value ± SEM (n = 3). (F) Same as E, except that the *ste11* mRNA was quantified. Each column represents the averaged value ± SEM (n = 3). (G) The occupancy of histone H3 at the *ste11* promoter was measured by ChIP using the indicated amplicon in the *tet0-snf21-TAP* strain grown in the presence of ahTet for two hours. Each column represents the averaged value ± SEM (n = 3).
The following figure supplement is available for figure 2:

**Figure supplement 1.** Subunits composition of the fission yeast SWI/SNF and RSC complexes - Schematic of the rTetR switch off system used.

increased of H3 occupancy, reminiscent of the effect of *lsk1* deletion (*Figure 2A*). Nucleosome scanning in the *arp9* deleted strain and the *rpb1 S2A* mutant confirmed these data and provided a moderate resolution analysis of the *ste11* promoter, demonstrating that the absence of either S2P or RSC results in higher nucleosome occupancy within the NDR of *ste11* (*Figure 3B*, *Figure 3—figure supplement 1*).

Taken together with our previous work (*Materne et al., 2015*), these data suggest that the H3 acetylation promoted by S2P is required for the recruitment of RSC and subsequent nucleosome eviction at the promoter of *ste11*. Supportive of this possibility, the abolition of either H3K14 acetylation or Lsk1 halved the occupancy of Rsc1-TAP at the *ste11* promoter (*Figure 3C–D*).

## The NuA3 and NuA4 histone acetyltransferases are not required for *ste11* transcription

Our current model of *ste11* regulation implies that H3K4me3 is a repressive mark that recruits the SET3C HDAC via the PHD domain of Set3. However, available data from other systems support that H3K4me3 rather correlates with high turnover nucleosomes near the transcription initiation site and this mark is recognized by the histone acetyltransferase complexes NuA3 and NuA4 via the PHD domain of the ING proteins Yng1 and Yng2 (Png1 and Png2 in fission yeast). We therefore tested the effect of inactivating NuA3 and NuA4 by combining a deletion of *mst2*, encoding the catalytic subunit of NuA3 (*Gomez et al., 2005*), with a *ts* allele of *mst1*, encoding the catalytic subunit of NuA3 (*Gomez et al., 2008*). The induction of *ste11* was barely affected in the absence of these two HATs (*Figure 3—figure supplement 2A*). This data render unlikely the possibility that NuA3 and NuA4 participate in *ste11* expression through H3K4me3 and support previous data showing that Gcn5 is the primary HAT acting at the *ste11* locus (*Helmlinger et al., 2008*). We also tested the effect of the H4K16R mutant (*Wang et al., 2012*), which behaved similarly to the wild type (*Figure 3—figure supplement 2B*).

## H2B ubiquitylation represses gametogenesis by opposing RSC dependent chromatin remodeling at the *ste11* promoter

Collectively, the previous experiments support that H2B ubiquitylation nearby the *ste11* promoter impedes proper chromatin remodeling by RSC through promoter histone deacetylation by SET3C, which represses transcription. The deletion of *rsc1* did not affect the level of H2B ubiquitylation nearby the *ste11* promoter but led to a decrease of the level of this modification in the transcribed region, most likely as a consequence of decreased transcription when Rsc1 is absent (*Figure 4A*). In order to further test this model, we compared the level of Arp9-TAP in the *htb1 K119R, hos2△ and lsk1△* strains to the wild type (*Figure 4B–D*). The Arp9 protein is a shared subunit of RSC and SWI/SNF but the latter is not present at the *ste11* locus (*Figure 2—figure supplement 1D*) and does not affect *ste11* expression (*Figure 2B*). The abolition of either H2B ubiquitylation or histone deacetylation both resulted in a marked increase of Arp9-TAP specifically at the promoter of *ste11* while the absence of S2P had the opposite effect. Moreover, histone H3 acetylation was increased at the *ste11* promoter when H2B ubiquitylation was abolished in either the *htb1 K119R* or the *rhp6* mutants

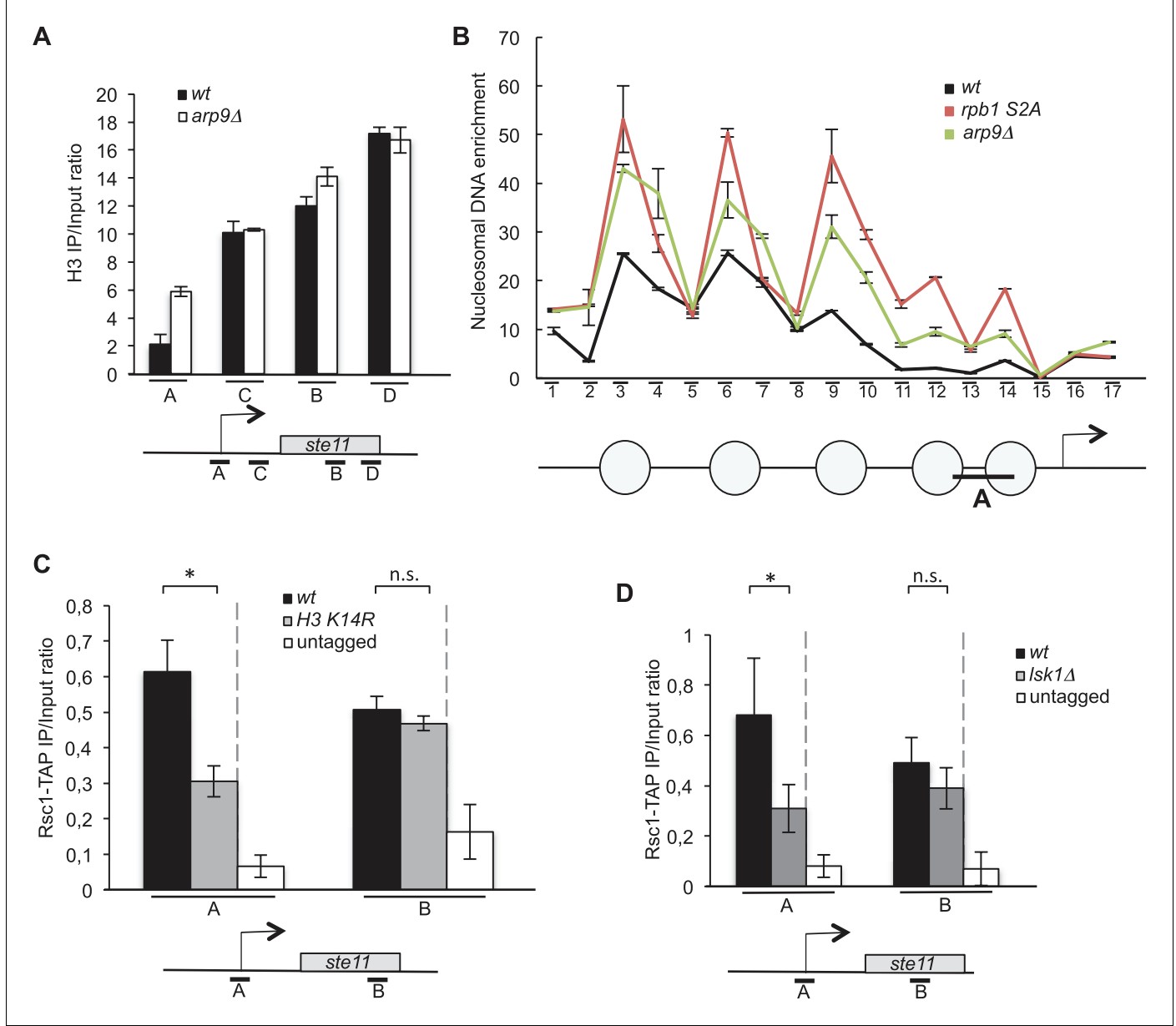

**Figure 3.** The RSC complex regulates nucleosomes occupancy at the *ste11* promoter. (**A**) The occupancy of histone H3 at the *ste11* locus was measured by ChIP using the indicated amplicons (**A–D**) in the indicated strains. Each column represents the averaged value ± SEM (n = 3). (**B**) Nucleosome scanning analysis of the indicated strains. Nucleosomal DNA enrichment at the indicated positions of the *ste11* locus was determined by ChIP experiment on MNase-digested chromatin. Data are presented as the average of three independent experiments along with the SEM. Inferred nucleosome locations are indicated. The bar indicates the position of amplicon A used in ChIP experiments. (**C**) The occupancy of Rsc1-TAP at the *ste11* locus was measured by ChIP using the indicated amplicons (**A–B**) in the *wt* and *htt2 K14R* strains. Each column represents the averaged value ± SEM (n = 4). To determine whether the decreased Rsc1-TAP enrichment was statistically significant, the difference in the means of enrichment between the ChIP peak in the wild type and the tested strains was estimated using t-test, assuming unequal variances between samples (Welch's t- test). *p-value* < 0.05 indicated by *, n.s. : non significant. (**D**) The occupancy of Rsc1-TAP at the *ste11* locus was measured by ChIP using the indicated amplicons (**A–B**) in the *wt* and *lsk1* deleted strains. Each column represents the averaged value ± SEM (n = 4). To determine whether the decreased Rsc1-TAP enrichment was statistically significant, the difference in the means of enrichment between the ChIP peak in the wild type and the tested strains was estimated using t-test, assuming unequal variances between samples (Welch's t- test). *p-value* < 0.05 indicated by *, n.s. : non significant.

The following figure supplements are available for figure 3:

**Figure supplement 1.** Schematic of amplicons used in the nucleosome scanning experiments.

*Figure 3 continued on next page*

*Figure 3 continued*

**Figure supplement 2.** The NuA3 and NuA4 HATs and histone H4 acetylation on K16 are not required for *ste11* induction.

(*Figure 4E*). Therefore, H2Bubi represses gametogenesis by opposing recruitment of the RSC remodeler at the *ste11* promoter.

## A genome-wide connection between RSC and CTD S2P

In order to test if the proposed connection between the phosphorylation of serine 2 within the CTD and the RSC remodeling complex could be expanded to more genes, we performed genome-wide mapping of nucleosome position by MNase-Seq in the *rsc1* deletion mutant. A meta-gene analysis of the nucleosome occupancy signal for all protein-coding genes revealed higher occupancy upstream of the TSS in the *rsc1* mutant, and an average 13 bp shift of the −1 nucleosome toward the TSS was noted (*Figure 4—figure supplement 1A–B*) when selecting the 10% protein-coding genes whose promoter nucleosome-depleted region (NDR) shrinks the most in the absence of Rsc1 (*Figure 4—figure supplement 1A*, right panel). The *ste11* gene belonged to that category (*Figure 4—figure supplement 1C*), confirming the previous single gene analyses performed above. We next analysed how of the previously established list of genes whose promoter nucleosome-depleted region (NDR) shrinks the most in the S2A mutant (*Materne et al., 2015*) behaved when *rsc1* is deleted (*Figure 4—figure supplement 1D*). Remarkably, these genes also showed higher nucleosome occupancy close to their TSS in the absence of Rsc1, indicating a mechanistic connection between the abolition of CTD S2P and the absence of Rsc1. Comparing the subset of genes whose promoter nucleosome-depleted region (NDR) shrinks the most in the absence of either Rsc1 or in the absence of S2P revealed a significant overlap (p=*3.416e-06*). These data support a genome-wide mechanistic connection between CTD phosphorylation on serine 2 and remodeling by the RSC complex.

## Discussion

Despite the fact that the phosphorylation of the CTD S2 is not essential in yeast, the master regulator of gametogenesis, *ste11*, requires an unusual requirement of S2P nearby its promoter for proper induction to counteract the repressed state imposed by the Set1-H3K4me3-HDAC pathway (*Materne et al., 2015*). A recent study proposed that Set1 represses the transcriptome independently of H3K4 methylation and is recruited by the Atf1 transcription factor (*Lorenz et al., 2014*). However, three previous studies have shown that Atf1 functions as an activator, rather than a repressor at the *ste11* locus (*Kanoh et al., 1996*; *Shiozaki and Russell, 1996*; *Takeda et al., 1995*) and a H3K4R mutant also derepresses *ste11* expression (*Materne et al., 2015*). Therefore, while we do not exclude that Set1 also represses *ste11* independently of its H3K4 methylase activity, this pathway is not predominant.

Both the deletion of *ubp8* (and therefore increased ubiquitylation) and the *htb1 K119R* mutant (and therefore absence of ubiquitylation) resulted in derepression of *ste11*. This is at first sight hard to explain as removal of histone modifications are expected to oppose the effect of their addition, such as in the acetylation/deacetylation process. However, a similar case was reported at the *GAL1* locus (*Henry et al., 2003*), already underlying the complex dynamic of H2Bubi where the sequential addition and removal of the mark is required for proper induction. Although the *ubp8* and *htb1 K119R* both result in elevated *ste11* expression, only the *htb1 K119R* suppresses the requirement of the S2 kinase Lsk1, which indicates that the positive effects of the addition and removal of the mark may operate through unrelated mechanisms. This is backed up by the spatially opposite behaviour of the H2Bubi mark over the *ste11* locus. The fact that the double *lsk1 ubp8* mutant behaves as the single *lsk1* mutant suggests that the function of *lsk1* is required upstream of *ubp8*. Indeed if Ubp8 positively affects elongation by facilitating the eviction of the H2A-H2B dimer and nucleosome reassembly in the wake of the polymerase (*Batta et al., 2011*; *Pavri et al., 2006*), it is expected that this effect will be masked by the strong defect in polymerase occupancy at the promoter observed when *lsk1* is absent. The positive effect of *ubp8* on elongation is likely to occur independently of H3K4 methylation as previously reported (*Tanny et al., 2007*) and maybe related to the positive role of

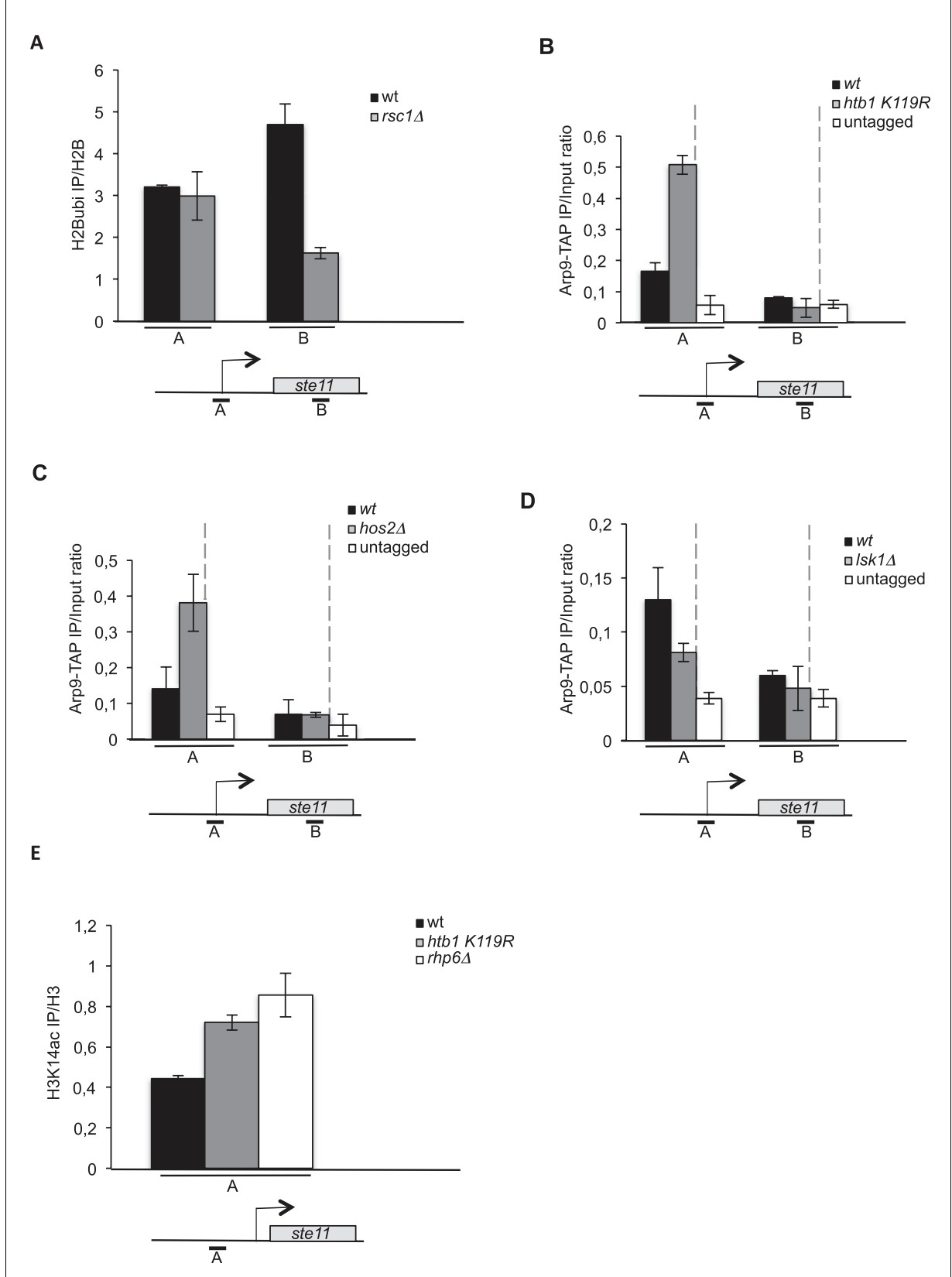

**Figure 4.** H2B-ubi and HDAC oppose the recruitment of the RSC complex at the ste11 promoter. (**A**) The occupancy of ubiquitylated H2B at the indicated locations (**A**, **B**) was determined by ChIP in the indicated strains using the anti-H2Bubi normalized against unmodified H2B. Each column

*Figure 4 continued on next page*

*Figure 4 continued*

represents the averaged value ± SEM (n = 3). (**B**) The occupancy of Arp9-TAP at the *ste11* locus was measured by ChIP using the indicated amplicons (A-B) in the *wt* and *htb1 K119R* strains. Each column represents the averaged value ± SEM (n = 3). (**C**) The occupancy of Arp9-TAP at the *ste11* locus was measured by ChIP using the indicated amplicons (**A–B**) in the *wt* and *hos2△* strains. Each column represents the averaged value ± SEM (n = 3). (**D**) The occupancy of Arp9-TAP at the *ste11* locus was measured by ChIP using the indicated amplicons (**A–B**) in the *wt* and *lsk1△* strains. Each column represents the averaged value ± SEM (n = 3). (**E**) The occupancy of acetylated H3K14 at the *ste11* promoter was determined by ChIP in the indicated strains using the anti-H3 K14ac normalized against unmodified H3. Each column represents the averaged value ± SEM (n = 3).

The following source data and figure supplements are available for figure 4:

**Figure supplement 1.** Genome-wide analysis of the absence of Rsc1 on NDR and connection between RSC and CTD S2P.

**Figure supplement 1—source data 1.** List of the 10% genes showing the strongest promoter NDR size decrease in the *rsc1* mutant (see *Figure 4—figure supplement 1A*, right panel) with their genomic coordinates, NDR size in wild type and the *rsc1* mutant.

**Figure supplement 2.** Schematic of the role of H2B-ub1 and RSC in the control of *ste11* expression.

H2Bubi in non-coding transcription at heterochromatic loci (*Sadeghi et al., 2014*; *Zofall and Grewal, 2007*).

The emerging model, complementing the recently propose role of S2P in *ste11* induction is that promoter nucleosome H2Bubi positively regulates the Set1 deposition of H3K4 methylation and therefore is the primary signal of HDAC-dependent repression of transcription (*Figure 4—figure supplement 2*). Upon transcriptional induction, the recruitment of SAGA-associated Ubp8 (*Helmlinger et al., 2008*) removes the ubiquitin mark, and concomitantly, the MAP kinase-dependent recruitment of Lsk1 deposits higher level of S2P. These two unrelated processes impede the Set1-dependent recruitment and/or activation of the SET3C HDAC, which provides access of the transcription machinery to the larger NDR that is formed. From the genetic analyses, as *ste11* is highly transcribed in the *ubp8* mutant, it appears that the S2P-dependent induction of transcription can bypass the requirement of de-ubiquitylation dependent on Ubp8.

Considering that H2B ubiquitylation was reported to act as a barrier to Ctk1 (the Lsk1 orthologue of budding yeast) (*Wyce et al., 2007*), we speculate that this additional layer of control (*Figure 4—figure supplement 2A*) may also participate in the negative role of H2B-ub1 on *ste11* induction. Supporting this possibility, the level of Lsk1 was increased at the *ste11* promoter when histone H2B ubiquitylation was abolished (*Figure 4—figure supplement 2B*).

The analysis of nucleosome occupancy indicates that RSC affects the NDR of *ste11* very similarly to S2P and indeed both S2P and histone acetylation are required for high occupancy of RSC at this locus. Considering that RSC recognizes acetylated nucleosomes through its bromodomains-containing subunits (*Carey et al., 2006*; *Kasten et al., 2004*), we propose that RSC functions as the ultimate player required to induce *ste11* transcription upon induction. Comparing the genome-wide nucleosome occupancy in the absence of either CTD S2P or *rsc1* reveals a significant overlap in the subset of genes whose NDR shrinks most, which constitutes an evidence for a mechanistic link between the phosphorylation of CTD serine 2 and the RSC complex.

Our data provide a mechanism by which promoter H2Bubi represses gene expression by opposing the recruitment of RSC. More specifically at the *ste11* locus, the balance between the negative effect of H2Bubi and the positive effect of RSC is critical for the timely induction of gametogenesis, which highlights the major role of chromatin regulation in cell fate. An important remaining issue is to understand why the H2Bubi-dependent H3K4 methylation only represses a subset of genes while this mark is a hallmark of most, if not all transcribed regions. Further work is in progress to address this point.

## Materials and methods

Fission yeast growth, gene targeting and mating were performed as described (*Bamps et al., 2004*; *Bauer and Hermand, 2012*; *Drogat et al., 2012*; *Fersht et al., 2007*). The expression of *ste11* was induced by nitrogen starvation. Western blot were performed with anti-H2B (Active Motif #39237), anti-H2Bubi (Active Motif #39623), PAP (Sigma #P1291) and anti-tubulin (Sigma #T5168) antibodies.

Iodine staining was performed as described (*Bauer et al., 2012*). The pREP-*cdc14* plasmid and the *cdc14-118 ts* mutant were gifts of Viesturs Simanis. Latrunculin A (LatA) was purchased from Enzo Life Sciences (#76343-93-6) and used at 0.5 µM. Anhydrotetracycline (AnTet) was purchased from Sigma (#37919) and used at 2.5 µg/ml. Plasmids used to generate the switch off *snf21* strain (*Zilio et al., 2012*) were a gift of Nicola Zilio.

## LatA sensitivity screen

The *S. pombe* deletion set was purchased from Bioneer. To screen for mutant phenotypes, the library was spotted onto YES and YES supplemented with 0.5 µM LatA. The 33 plates were manually scored for growth both 2 and 5 days after incubation. The screen was performed two times with the entire set, and potential hits were retested. The complete list of genes whose deletion results in LatA sensitivity will be published elsewhere.

## ChIP and quantitative RT-PCR

Chromatin Immunoprecipitations were performed using a Bioruptor (Diagenode) and Dynabeads (Invitrogen). Precipitated DNA was purified on Qiagen. Quantitative RT-PCR was performed using the ABI high capacity RNA-to-cDNA (*Devos et al., 2015*). The untreated sample was used as a reference and the *act1* mRNA was used for normalization. Antibodies used in ChIP were anti-H2B (Active Motif #39237), anti-H2Bubi (Active Motif #39623), anti-H3K4me3 (Millipore #07–473), anti-H3K14ac (Millipore #07–353) anti-H3 (Abcam #1791) and PAP (Sigma). For all ChIP experiments, each column represents the mean percentage immunoprecipitation value ± SEM (n = 2–4). Note that a 10 min crosslinking in formaldehyde was used as routine but was extended to 15 min in the case of Rsc1-TAP.

## Nucleosome scanning

A culture of 500 ml of fission yeast cells was grown to OD 0.5 at 32°C and crosslinked with 7 ml of Formaldehyde 37% for 20 min at 25°C, 60 rpm. The crosslink was stopped by the addition of 27 ml of Glycine and cells were pelleted. The pellet was resuspended in preincubation solution (Citric acid 20 mM, Na2HPO4 20 mM, EDTA pH 8 40 mM) supplemented with 100 µl β mercaptoethanol/50ml and incubated 10 min at 30°C. The cells were centrifugated and resuspended in 10 ml of Sorbitol 1 M / Tris pH 7.4 50 mM buffer containing 200 µl Zymolase (0,01g/200 µl water) and incubated 20 min at 30°C (40 min when EMM medium was used). After centrifugation, the pellet was resuspended in 7.5 ml NP buffer (Sorbitol 1 M, NaCl 50 mM, Tris pH 7.4, 10 mM, MgCl$_2$ 5 mM, CaCl$_2$ 1 mM, NP-40 0.75%) supplemented with 7.6 µl NP buffer + 0,5 µl β mercaptoethanol + 400 µl spermidine 10 mM) and split into 2 Falcon tubes (Total and MNase treated). Add 50 µl MNase (32 units) to one tube and incubate 20 min at 37°C without agitation. Add 500 µl Stop buffer, 200 µl RNase A (0.4 mg/ml) and 225 µl proteinase K (20 mg/ml) and incubate at 65°C overnight. Potassium Acetate was added (1.25 ml of a 3M solution) and the mix was incubated 5 min on ice. After phenol extraction, 200 µl NaCl 5 M, 1.7 µl Glycogene (20 mg/ml), and 3.5 ml of isopropanol were added. After precipitation and ethanol wash, the pellet was resuspended in 200 µl of TE buffer. The samples were run on agarose gel (1.5%) and the bands corresponding to the mononucleosomes were cut and purify with Qiagen. Q-PCR with the primer pairs of the set of overlapping amplicons were performed. Nucleosomal DNA enrichment calculated as the ratio between the amounts of PCR product obtained from DNA samples generated from the mononucleosomal gel purification to that of the input (total) DNA.

## RT-Q-PCR

Total RNA was prepared as described (*Guiguen et al., 2007*) and purified on Qiagen RNeasy. Q-RT-PCR was performed using the ABI high capacity RNA-to-cDNA following the instructions of the manufacturer. The untreated sample was used as a reference and the *act1* mRNA was used for normalization. In all Q-RT-PCR experiments, each column represents the averaged value ± SEM (n = 2–3).

## MNase-Seq

The preparation of mononucleosomal DNA and the sequencing of mononucleosomal DNA were previously described in details (*Soriano et al., 2013*). The nucleosome sequencing data have been deposited in the GEO database under the accession number GSE80524.

MNase-seq data was processed and dynamic changes were detected using DANPOS (Dynamic Analysis of Nucleosome Positioning and Occupancy by Sequencing - https://code.google.com/p/danpos/). Clonal reads (determined by their very high coverage compared to the mean coverage across the genome based on a Poisson P-value cutoff) were removed from the reads previously mapped on the *S.pombe* genome with BWA (bio-bwa.sourceforge.net). Variation in size resulting from MNase treatment were compensated by shifting each read toward the 3' direction for half of the estimated fragment size. Nucleosome occupancy was then calculated as the quantile-normalized count of adjusted reads covering each base pair in the genome. Afer this processing, DANPOS calculates the differential signal at single nucleotide position based on a Poisson test. Dynamic nucleosomes are then identified by peak calling on these signals. TSS associated NDR lengths were quantified as the length of the longest DNA segment whose proximal border is located closer than 65 bp from the TSS with nucleosome occupancy levels lower than an arbitrary treshold (mean(occupancy$_{genome}$) – standard_deviation(occupancy$_{genome}$)) at any point.

The processed data can be visualized on the web browser http://genomics.usal.es/Materne2016.

## Acknowledgements

We thank the GEMO laboratory for discussions. We thank Susan Forsburg, Jason Tanny, Nicola Zilio and Viesturs Simanis for strains. We thank Dominique Helmlinger and Jason Tanny for critical reading of the manuscript. This work was supported by grant BFU2014-52143-P from the Spanish Ministerio de Economía y Competitividad to FA and by grants PR T.0012.14, MIS F.4523.11, Ceruna and Marie Curie Action to DH. DH is a FNRS Senior Research Associate.

## Additional information

### Funding

| Funder | Grant reference number | Author |
|---|---|---|
| The Spanish Ministerio de Economia y Competividad | BFU2014-52143-P | Francisco Antequera |
| Fonds National de la Recherche Scientifique | PR T.0012.14 | Damien Hermand |

The funders had no role in study design, data collection and interpretation, or the decision to submit the work for publication.

### Author contributions

PM, EV, MS, JA, VM, Acquisition of data, Analysis and interpretation of data; CY-S, Analysis and interpretation of data, Drafting or revising the article; FA, Conception and design, Analysis and interpretation of data; DH, Conception and design, Analysis and interpretation of data, Drafting or revising the article

### Author ORCIDs

Damien Hermand, http://orcid.org/0000-0002-1029-5848

## Additional files

### Supplementary files

• Supplementary file 1. The file contains a list of oligonucleotides and strains used in the study.

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
