## [Decision Letter]

Thank you for submitting your work entitled "Histone H2B ubiquitylation represses gametogenesis by opposing RSC-dependent chromatin remodeling at the *ste11* locus" for consideration by *eLife*. Your article has been reviewed by three peer reviewers, and the evaluation has been overseen by Detlef Weigel as the Senior Editor and Reviewing Editor.

The reviewers have discussed the reviews with one another and the Reviewing Editor has drafted this decision to help you prepare a revised submission.

The reviewers liked that this work is beginning to answer some of the questions left open in your original *eLife* paper. However, they also felt that you needed to go beyond investigating *ste11* as a model locus to broaden the general appeal of the work, by incorporating new MNase and ChIP-seq data sets. There are other examples where the H3K4me3 has been shown to be repressive because it recruits a complex containing HDAC activity and a PHD-containing subunit that reads the histone mark. For example, at PHO5, H3K4me3 is both positive during activation in S/G2/M and repressive during mitotic exit when the Rpd3L HDAC is recruited via two PHD-containing subunits. The step forward in your paper is the identification of RSC in the genetic screen and its placement at the end of the pathway identified in the previous *eLife* paper. It could be argued, though, that reduced acetylation would be expected to impair recruitment of RSC. Given that one could question how big a step the present manuscript is, I would also support requesting the genome-wide datasets to broaden general appeal beyond *ste11*.

*Reviewer #1:*

This paper by the Hermand lab uses LatA sensitivity screen to identify potential factors controlling gene expression together with RNAPol II Ser2 phosphorylation (S2P). H2B ubiquitin protease Ubp8 and some components of RSC and SWI/SNF complex were identified, which suggests potential link between H2B-ub1/chromatin remodeling and S2P pathway. In fission yeast gametogenesis, *ste11* is a key transcriptional master regulator. The authors focus on *ste11* transcription, as their previous recent paper showed that *ste11* is regulated by S2P and by Set1 K4me-repression. Here they show that *ste11* is repressed by H2Bub and activated by RSC. It is proposed that, via Set1-H3K4me3-HDAC pathway, H2Bub lowers RSC recruitment to ste11 promoter and regulates local nucleosomal occupancy and therefore transcription. These results may explain how H2B-ub at the promoter represses *ste11* transcription.

The overall findings are interesting with respect to the master regulatory *ste11* in fission yeast. Most interesting is that the typically activating chromatin modification, H3K4me3, is thought to be repressive via recruitment of HDAC complex and, in the new data here, resisting Rsc remodeling of promoter. In light of the previous paper, which made the basic observation of this pathway regulating *ste11*, there are two key questions for this to increase broad interest:

1) Are there general mechanisms to control gene expression in fission yeast, which can be addressed by genome wide studies in addition to specific analyses of *ste11*?

2) Is there an effect on gametogenesis itself, not only on expression of *ste11*?

Additional specific technical questions:

1) Since H2Bub and RSC are both regulating *ste11* expression in fission yeast, do RSC mutants affect H2Bub levels at *ste11* promoter?

2) KR mutagenesis is not perfect to mimic non-ubiquitylated lysine, which may cause side effects. Authors should test the rhp6 deletion strain in at least one key experiment, e.g. in Figure 1 and Figure 1—figure supplement 1.

3) Results: 'We conclude that H2B ubiquitylation at the promoter of *ste11* represses transcription by increasing H3K4-3me', There is no evidence to support "by increasing H3K4-3me".

4) Authors conclude that H2Bub opposes RSC to remodel chromatin at *ste11* promoter via Set1-H3K4me3-SET3C HDAC pathway. So does KR mutant and rhp6 deletion affect H3K14ac at *ste11* promoter.

5) Quantification: no p-values – need to be added. How many biological repeats were performed for each experiment? I could not find this in Figure Legends.

6) In Figure 1 and elsewhere, the ChIP signals should be normalized to histones, rather than input, since the authors are arguing that levels of histones are changing.

7) In Figure 1—figure supplement 1: To make the statement that H2Bub is increased in ubp8 deleted strains, the blots should be quantified and the level of H2Bub needs to be normalized to H2B.

8) Figure 2: Since viability is low at 3 days (D), it is important to show viability test at timing of the transcription experiment in E.

9) Figure 3: Since nucleosome occupancy is increasing everywhere in the promoter with the mutations in Rsc, the question is how specific is this? What about the gene body? What about other genes and promoters?

*Reviewer #2:*

This manuscript follows up on studies published earlier in *eLife* by the authors. In that study, they presented genome-wide nucleosome mapping data in fission yeast, *S. pombe*, showing that the absence of phosphorylation of the CTD of RNA polymerase II (RNAP II) on serine 2 (S2) led to increased nucleosome occupancy and repositioning in the promoters of a subset of yeast genes. The affected promoters generally had a larger nucleosome depleted region (NDR) than the majority of genes, and included the *ste11* gene, which the authors previously found had a peak of S2 phosporylation (S2P) near the promoter during its activation. They showed that the peak of S2P at *ste11* and a second promoter reduced nucleosome occupancy and increased histone acetylation during gene activation. The take home message from these and other studies in this first manuscript was that during activation S2P counteracts repressive histone deacetylation by SET3C, which is targeted to promoters via its interaction with RNAP II-S5P/Set1-mediated H3K4 methylated histones. The dual presence of S2P and S5P was also found to block Set1 interaction with the CTD.

In the present manuscript, the authors focused on the *ste11* gene to investigate the role of other regulators in this pathway of promoter repression. Their examination was restricted to H2B monoubiquitylation (H2B-ub1) and the RSC nucleosome-remodeling complex, which were identified in a screen for mutants that relieved the drug sensitivity of lsk1, the fission yeast S2 CTD kinase. By analyzing loss of function mutations coupled with RNA analysis and ChIP, the authors report that an htb1KR mutation suppressed the transcription defect of a lsk1 deletion and that H2B-ub1 was localized to the *ste11* promoter under basal (repressed) conditions. As expected, its promoter occupancy positively correlated with the occupancy of H3K4me3. Consistent with a role in promoter repression through its trans-regulation of H3K4me3, an htb1-KR mutant showed reduced nucleosome occupancy in the ste11 promoter. In contrast, mutations that decreased RSC expression had the opposite phenotype – *ste11* transcription was reduced and nucleosome occupancy was increased in the promoter. The authors propose that H2B-ub1 represses transcription of *ste11* by opposing RSC-dependent nucleosome remodeling.

The phenocopying of S2A and H3K4me3 mutants with rsc and htb-K119R mutants, respectively, supports the placement of Rsc and H2B-ub1 in the pathway leading to *ste11* activation, thus extending the author's initial definition of this pathway and adding some interesting new information about the interplay between RNAP II phosphorylation and chromatin remodeling at promoters. However, gaps in the analysis remain and should be addressed, as outlined below.

1) It is unclear why RSC mutants were identified in the screen for suppression of drug sensitivity of Lsk1. The authors should examine the effects of a rsc1∆ lsk1∆ double mutation on *ste11* transcription as they did for an htb-K119R lsk1∆ double mutation.

2) The authors used Arp9-TAP to perform ChIP experiments in Figure 4. Because Arp9 is present in both RSC and SWI/SNF complexes, they should confirm that the results were due only to the recruitment of RSC. Similarly, the authors should show the effect of rsc1∆ as well as arp9∆ on H3 levels in the *ste11* promoter.

3) Nucleosome scanning should be included in the analysis of the phenotypes of the htb-K119R mutant.

4) An important missing gap in the pathway is the relationship of H2B-ub1 to S2-P; for example, does H2B-ub1 affect Lsk1 recruitment to the promoter and subsequent S2-P? Along the same lines, what is responsible for the deposition of H2B-ub1 at the *ste11* promoter, for example what is the role of S5-P, and what is the status of Bre1 occupancy at the promoter?

5) Present evidence confirming that the htb1-KR mutant does not show HDAC recruitment to the *ste11* promoter and that H3ac levels remain high, consistent with H2Bub1 being upstream of H3K4me3 in the pathway.

6) This study would be significantly enhanced if the authors extended their MNase mapping experiments genome wide in rsc∆ mutants, and then comparing the results to those reported for an S2A mutant. This would broaden the authors' conclusions and complement their in-depth analysis of a single gene.

*Reviewer #3:*

Previously, Materne et al. reported (*eLife*. 2015. 4:e09008) that phosphorylation of the RNA polymerase II CTD on Ser2 (S2P) controlled nucleosome dynamics in *S. pombe* at the promoters of 324 genes, including *ste11* that encodes a regulator of mating and gametogenesis. The authors also found that MAP kinase activation of latrunculin-sensitive kinase 1 (Lsk1) mediated the S2P modification. Mechanistically, the authors established that CTD S2P by Lsk1 counters CTD S5P-dependent recruitment of Set1-mediated trimethylation of histone H3 lysine 4 (H3K4me3), which, in turn, recruits the SET3C histone deacetylase (HDAC) complex.

In the present submission, Materne et al. identified additional regulators of *ste11* expression, and cdc14 to a lesser extent, using a genetic screen for genes required to complete cytokinesis in the presence of lantruculin A (LatA), a disruptor of the actinomyosin ring. The screen identified SAGA-associated Ubp8, a ubiquitin (ub) protease that removes monoubiquitin from H2B, as well as shared and non-shared subunits of the chromatin remodelers RSC and SWI/SNF (Figure 1—figure supplement 1). The identification of Ubp8 is not unexpected as it is well documented that H2Bub is required for trans-modification of H3 and promoter accumulation of H3K4me3, which recruits SET3C HDAC activity to *ste11*. Nevertheless, using ChIP (using a newly available monoclonal antibody) the authors convincingly show that the level of H2Bub1 at the *ste11* correlates with the presence of H3K4me3, and both histone marks decrease on activation (Figure 1). Consistent with this, a H2BK119R mutation, which abolishes H2Bub1 and hence H3K4me3 at the *ste11* promoter (Figure 1), decreased H3 occupancy (Figure 2) and increased both basal and induced levels of ste11 expression (Figure 1). Loss of SWI/SNF and RSC subunits demonstrated that RSC, but not SWI/SNF, is required for nucleosome depletion of the *ste11* promoter (Figure 3) and its activation (Figure 2). As shown by ChIP, RSC is recruited specifically to the *ste11* promoter as opposed to the gene body, and this recruitment increases when H2Bub1 is abolished by H2BK119R (Figure 4). Conversely, RSC recruitment is decreased in the absence of H3K14ac (H3K14R mutant) and lsk1∆ cells, which lack S2P that is needed to inhibit H2B ubiquitylation (Figure 3). Genetic (Figure 1) and RT-Q-PCR (Figure 2—figure supplement 1, evidence is also presented indicating that Cdc14 is a downstream target of Lsk1 and thus requires S2P as well as the remodelers Rsc and SWI/SNF for activation.

The manuscript is written very well and understandably. In addition, all of the experiments are executed well and the data support the authors' conclusions and model (Figure 4), whereby H2Bub1, by increasing H3K4me3, recruits the SET3C HDAC and thus antagonizes recruitment of RSC and activation of *ste11*. The manuscript clearly builds on their previous paper and is suitable for publication in as an *eLife* Research Advance.

Figure 3: The conclusion that "high promoter occupancy of Rsc1-TAP required both H3 K14 acetylation and the S2 kinase Lsk1" is unsupported. ChIP only reports relative occupancy and there is a two-fold decrease in RSC occupancy in the H3K14R and isk1delta strains. The wording should be changed to something to the effect of the mutants halved the occupancy of RSC or significantly decreased occupancy. The latter raises the point of whether the reduction in occupancy in both panels is statistically significant.

Introduction, first paragraph: "However, in the absence of Set1, most genes are upregulated, suggesting a predominant role in repression." To the best of my knowledge, this is incorrect. In budding yeast, Boa et al. (Yeast. 2003. 20, 827) reported that transcript levels of most genes (80%) were downregulated in the absence of Set1 and H3K4 methylation. In contrast, Set1 and H3K4 methylation does silence the rDNA (Briggs et al. 2001 Gene Dev 15, 3286). The statement should be justified or corrected and referenced appropriately.

---

## [Author Response]

*Reviewer #1:*

1) Are there general mechanisms to control gene expression in fission yeast, which can be addressed by genome wide studies in addition to specific analyses of ste11?

We have addressed the question by analysing the genome wide effect of the deletion of rsc1, a non-essential subunit of RSC, on nucleosome positioning, and by comparing it with the CTD S2A mutant. We decided to perform an MNase-Seq experiment, which is time consuming but provides a high resolution map of nucleosome. A new figure (Figure 4—figure supplement 1) with 5 panels is now presented and the corresponding text reads as follows:

“In order to test if the proposed connection between the phosphorylation of serine 2 within the CTD and the RSC remodeling complex could be expanded to more genes, we performed genome-wide mapping of nucleosome position by MNase-Seq in the rsc1 deletion mutant. […] These data support a genome-wide mechanistic connection between CTD phosphorylation on serine 2 and remodeling by the RSC complex.”

To answer the question of the referee, we can therefore conclude that the mechanism we describe in details in the case of *ste11* can likely be extended to a subset of genes (82 when considering the most stringent analyses) in the conditions we have tested (growth in standard conditions).

*2) Is there an effect on gametogenesis itself, not only on expression of ste11?* The referee asks the important question of the biological relevance of our analyses. We have added a panel in Figure 2—figure supplement 1 showing that similarly to the CTD S2A mutant, an rsc1 mutant also displays a prominent defect during gametogenesis while a snf22 mutant (encoding a subunit of the SWI/SNF complex) does not. Considering that the deletion of rsc1 affects *ste11* expression (Figure 2) while the deletion of snf22 does not (Figure 2), the molecular data are nicely supported by the physiological data. The text was modified as follows:

“The absence of either rsc1 or arp9 clearly affected the expression of *ste11* while snf22 had no effect (Figure 2), pointing to a more prominent role of RSC for *ste11* expression, which was confirmed by a gametogenesis assay (Figure 2—figure supplement 1).”

Additional specific technical questions:

1) Since H2Bub and RSC are both regulating ste11 expression in fission yeast, do RSC mutants affect H2Bub levels at ste11 promoter?

In order to answer that question, we have performed a ChIP experiment presented in Figure 4. We have chipped H2Bubi (normalized on a H2B ChIP) in a wild type and a rsc1 deleted strain. The absence of rsc1 does not seem to affect the level of ubiquitylation at the promoter of *ste11* but leads to a strong decrease of ubiquitylation in the 3’ region. This is not surprising considering that *ste11* transcription is strongly impeded in the absence of Rsc1 as judged by Q-RT-PCR (Figure 2). The text was modified as follows:

“The deletion of rsc1 did not affect the level of H2B ubiquitylation nearby the ste11 promoter but led to a decrease of the level of this modification in the transcribed region, most likely as a consequence of decreased transcription when Rsc1 is absent (Figure 4).”

2) KR mutagenesis is not perfect to mimic non-ubiquitylated lysine, which may cause side effect. Authors should test the rhp6 deletion strain in at least one key experiment, e.g. in Figure 1 and Figure 1—figure supplement 1.

We agree with the referee that the KR is not a perfect mimic of the non-ubiquitylated lysine. The referee may have missed that we had already performed the experiment he requested as shown in Figure 1—figure supplement 1 where the effect of the KR mutant was compared to a deletion of rhp6. We have also added a new experiment in Figure 4 where the level an acetylation on histone H3 lysine 14 is compared in a wild type strain, the KR mutant and the rhp6 deletion. In both experiments, the deletion of rhp6 and the KR mutant resulted in similar defects. We are therefore confident that in the case we are studying, the KR mutant behaves as a good mimic of the non-ubiquitylated state (see also point 4 below).

3) Results: 'We conclude that H2B ubiquitylation at the promoter of ste11 represses transcription by increasing H3K4-3me', There is no evidence to support "by increasing H3K4-3me".

The data presented in Figure 1 show a clear dependency of H3K4 methylation to H2B ubiquitylation, confirming the model generally admitted in the literature. However, we agree that the sentence stated by the referee is overstating the data. We have therefore changed it to:

“We conclude that H2B ubiquitylation at the promoter of *ste11* correlates with repressed transcription and high level of H3K4me3, which in turn may favour histone deactetylation by the SET3C complex.”

4) Authors conclude that H2Bub opposes RSC to remodel chromatin at ste11 promoter via Set1-H3K4me3-SET3C HDAC pathway. So does KR mutant and rhp6 deletion affect H3K14ac at ste11 promoter.

As mentioned above (point 2), we have tested the issue raised by the referee by chipping the K14 acetylated histone H3 (normalized on total H3) in the strains lacking H2B ubiquitylation, which results an elevated acetylation at the *ste11* promoter. The text was modified as follows:

“Moreover, histone H3 acetylation was increased at the *ste11* promoter when H2B ubiquitylation was abolished in either the htb1-K119R or the rhp6 mutant (Figure 4).”

5) Quantification: no p-values – need to be added. How many biological repeats were performed for each experiment? I could not find this in Figure Legends.

The number of replicates that was previously indicated in the Materials and methods is now mentioned for each experiment in the figures legend, as requested by the referee.

When a statistical analysis was performed as in Figure 3 and Figure 4—figure supplement 1 (right panel), the test used is indicated in the figure legend with the corresponding p-value.

6) In Figure 1 and elsewhere, the ChIP signals should be normalized to histones, rather than input, since the authors are arguing that levels of histones are changing.

In all experiments where the occupancy of modified histones was assessed, the signal was indeed normalized to the total corresponding histone. This is now more clearly indicated in the y-axis of the corresponding graphs.

7) In Figure 1—figure supplement 1: to make the statement that H2Bub is increased in ubp8 deleted strains, the blots should be quantified and the level of H2Bub needs to be normalized to H2B.

We agree with the referee. The scanned blots were quantified in ImageJ and the level of H2Bubi was normalized to H2B with the wild type set as 1.

8) Figure 2: Since viability is low at 3 days (D), it is important to show viability test at timing of the transcription experiment in E.

The timing of these experiments is very different. As the snf21 gene is essential, it was important to show that the switch-off system we used is sufficiently efficient to ultimately mimic a deletion. Indeed, the strain does not sustain growth when the promoter is turned off (Figure 2).

In the experiments presented in Figure 2, the timing is very short (three hours) in order to avoid as much as possible secondary effects. Within this time scale, corresponding to roughly one cell cycle, we have not noticed any effect on cell number. To us, this is not surprising as the depletion of snf21 is not yet complete and effect of cellular growth likely take longer to be apparent. We decided to use this time scale to ensure that the effects we observed are likely not a secondary consequence of a global misregulation of nucleosome positioning.

9) Figure 3: Since nucleosome occupancy is increasing everywhere in the promoter with the mutations in Rsc, the question is how specific is this? What about the gene body? What about other genes and promoters?

The nucleosome scanning experiment presented in Figure 3 indeed focuses on the promoter region of *ste11*. It is difficult to extend it further as the data presented already required more than 30 primers. However, the ChIP experiment presented in Figure 3 covers the entire locus (at a lower resolution) and clearly demonstrates that the occupancy effect occurs at the promoter and not on the gene body. Regarding other genes and promoters, we have now performed a genome-wide analysis of occupancy as presented in Figure 4—figure supplement 1.

We hope that these comments answer the issues raised by this referee and we thank him for his deep analysis of our work.

Reviewer #2:

1) It is unclear why RSC mutants were identified in the screen for suppression of drug sensitivity of Lsk1. The authors should examine the effects of a rsc1∆ lsk1∆ double mutation on ste11 transcription as they did for an htb-K119R lsk1∆ double mutation.

The referee is surprised that subunits of RSC were identified in the latA screen presented in Figure 1 and Figure 1—figure supplement 1. We would like to highlight that, contrary to what the referee wrote, this is not a screen for suppression of drug sensitivity but rather a screen for sensitivity to latA. The logic of the screen is that the CTD S2 kinase Lsk1 (latrunculin sensitive kinase) was shown to be sensitive to latA, most likely because the expression of one or several genes required for survival in the presence of the drug are affected. Our data support that the cdc14 gene encodes such a target gene. If Lsk1 and RSC are required in the same pathway to enforce cdc14 expression, it is not surprising to us that subunits of RSC were identified in the screen, as their deletion results in lower gene transcription. In the case of cdc14, contrary to *ste11*, both RSC and SWI-SNF seem to be required for proper expression.

Regarding the experiment proposed by the referee, namely to examine the effect of a double rsc1 lsk1 mutant, the combined deletion of these two genes is synthetic lethal as reported in our previous work (Materne et al., *eLife* 2015). Therefore, the experiment cannot be performed.

2) The authors used Arp9-TAP to perform ChIP experiments in Figure 4. Because Arp9 is present in both RSC and SWI/SNF complexes, they should confirm that the results were due only to the recruitment of RSC. Similarly, the authors should show the effect of rsc1∆ as well as arp9∆ on H3 levels in the ste11 promoter.

While the work was in progress, we noticed that the Rsc1 protein was difficult to Chip, resulting in less reproducible results. We therefore had to increase the timing of crosslinking as indicated in the Materials and methods. The Arp9 protein was easier to work with. As pointed by the referee, this raised the question of the specificity. Below is a list of the evidences supporting a specific role of RSC at the *ste11* promoter:

Figure 2: the effect of the deletion mutants of rsc1, snf22 and arp9 on *ste11* expression are presented, which points to a specificity of RSC

Figure 2: the transcriptional repression of the expression which snf21, which encodes the catalytic subunit of RSC, is sufficient to decrease *ste11* expression.

Figure 2—figure supplement 1: the deletion of rsc1, but not snf22 leads to a gametogenesis defect. Figure 2—figure supplement 1: the SWI-SNF specific subunit, Snf22, does not Chip at the *ste11* promoter, while Rsc1 and Arp9 does (Figure 3, Figure 4).

Moreover, we have performed a new experiment (Figure 2), showing that depletion of Snf21rapidly results in higher H3 level at the *ste11* promoter.

We hope that, taken together, these data convince the referee of a specific role of RSC at the *ste11* promoter.

3) Nucleosome scanning should be included in the analysis of the phenotypes of the htb-K119R mutant.

We have performed the requested experiment, which is difficult in this case due to the slow growth of the mutant. This has technical implications as the time required to digest the cell with zymolyase is critical and affected by many parameters, including the physiological state of the cell. The result is presented below and shows that the H2B K119R mutant behaves very similarly to a wild type, or could even have lower occupancy.

Author response image 1.**DOI:**
http://dx.doi.org/10.7554/eLife.13500.015

4) An important missing gap in the pathway is the relationship of H2B-ub1 to S2-P; for example, does H2B-ub1 affect Lsk1 recruitment to the promoter and subsequent S2-P? Along the same lines, what is responsible for the deposition of H2B-ub1 at the ste11 promoter, for example what is the role of S5-P, and what is the status of Bre1 occupancy at the promoter?

The referee points to an important issue, namely the relationship of H2Bubi to S2P. Indeed, previous work from the Berger lab has revealed that H2Bubi acts as a barrier to the S2 kinase nucleosomal recruitment, which we have mentioned in the Discussion. In order to test the effect of H2Bubi on Lsk1 recruitment, we have performed a ChIP experiment of Lsk1 in a wild type and a H2B K119R strains (Figure 4—figure supplement 2). The result indicates that the absence of ubiquitylation on H2B correlates with higher level of Lsk1 at the *ste11* promoter, which supports the model previously reported. The text was modified as follows:

“Considering that H2B ubiquitylation was reported to act as a barrier to Ctk1 (the Lsk1 orthologue of budding yeast) (Wyce et al., 2007), we speculate that this additional layer of control (Figure 4—figure supplement 2) may also participate in the negative role of H2B-ub1 on *ste11* induction.Supporting this possibility, the level of Lsk1 was increased at the *ste11* promoter when histone H2B ubiquitylation was abolished (Figure 4—figure supplement 2).”

The referee then asks additional questions regarding the regulation (deposition, status of Bre1) of H2Bubi. Some aspects have been largely covered over the years by previous studies from the Tanny, Gould, Ekwall, Grewal, Winston and Allis laboratories, leading to a pretty good description of the pathway leading the H2B ubiquitylation in fission yeast. In addition, we believe that these experiments are beyond the scope of the current work and message. Moreover, they would require several months of work. For example, we have no experience with Bre1 ChIP and preliminary set up for these type of experiments is always time consuming.

5) Present evidence confirming that the htb1-KR mutant does not show HDAC recruitment to the ste11 promoter and that H3ac levels remain high, consistent with H2Bub1 being upstream of H3K4me3 in the pathway.

We present a new experiment where we have tested the issue raised by the referee by chipping the acetylated K14 histone H3 (normalized on total H3) in the strains lacking H2B ubiquitylation, which results an elevated acetylation at the *ste11* promoter. This constitutes an evidence that H2B-ubi is upstream of the repressive deacetylation occurring at the *ste11* promoter. The text was modified as follows:

“Moreover, histone H3 acetylation was increased at the *ste11* promoter when H2B ubiquitylation was abolished in either the htb1-K119R or the rhp6 mutant (Figure 4).”

*6) This study would be significantly enhanced if the authors extended their MNase mapping experiments genome wide in rsc∆ mutants, and then comparing the results to those reported for an S2A mutant. This would broaden the authors' conclusions and complement their in-depth analysis of a single gene.* We have responded to this request by analysing the genome wide effect of the deletion of Rsc1, a non essential subunit of RSC, on nucleosome positioning, and by comparing it with the CTD S2A mutant. A new figure (Figure 4—figure supplement 1) with 5 panels is now presented and the corresponding text reads as follows:

“In order to test if the proposed connection between the phosphorylation of serine 2 within the CTD and the RSC remodeling complex could be expanded to more genes, we performed genome-wide mapping of nucleosome position by MNase-Seq in the rsc1 deletion mutant. […] These data support a genome-wide mechanistic connection between CTD phosphorylation on serine 2 and remodeling by the RSC complex.”

This new large-scale analysis constitutes an evidence that the mechanism we describe in details in the case of *ste11* can be extended to a subset of genes.

*Reviewer #3: Figure 3: The conclusion that "high promoter occupancy of Rsc1-TAP required both H3 K14 acetylation and the S2 kinase Lsk1" is unsupported. ChIP only reports relative occupancy and there is a two-fold decrease in RSC occupancy in the H3K14R and isk1*∆ *strains. The wording should be changed to something to the effect of the mutants halved the occupancy of RSC or significantly decreased occupancy. The latter raises the point of whether the reduction in occupancy in both panels is statistically significant.*

The sentence has been changed to:

“Supportive of this possibility, the abolition of either H3K14 acetylation or Lsk1 halved the occupancy of Rsc1-TAP at the *ste11* promoter.”

To determine whether the decreased Rsc1-TAP enrichment was statistically significant, the difference in the means of enrichment between the ChIP peak in the wild type and the tested strains was estimated using t-test, assuming unequal variances between samples (Welch's t- test). p-value < 0.05 indicated by *, n.s.: non-significant. This is added to the legend.

Introduction, first paragraph: "However, in the absence of Set1, most genes are upregulated, suggesting a predominant role in repression." To the best of my knowledge, this is incorrect. In budding yeast, Boa et al. (Yeast. 2003. 20, 827) reported that transcript levels of most genes (80%) were downregulated in the absence of Set1 and H3K4 methylation. In contrast, Set1 and H3K4 methylation does silence the rDNA (Briggs et al. 2001 Gene Dev 15, 3286). The statement should be justified or corrected and referenced appropriately.

The referee cites works done in budding yeast. In fission yeast, to our knowledge, two studies have analysed the effect of set1 deletion on gene expression genome-wide (Tanny et al. Genes Dev. 2007 Apr 1;21(7):835-47) and Lorenz et al. *eLife*. 2014 Dec 15;3:e04506). They both revealed a predominantly repressive role of Set1.

In order to avoid any ambiguity, we have added “in fission yeast” in the sentence and also added the proper reference.

In addition, we now mention two works done in budding yeast that supports a gene-specific repressive role of Set1.

In budding yeast, Set1 is the only H3K4 methyltransferase and plays a repressive role on PHO5, PHO84 and GAL1 expression, suggesting that H3K4me creates a repressive chromatin configuration (Carvin and Kladde, 2004; Wang et al., 2011)